# OpenReview forum: "EigenBench: A Comparative Behavioral Measure of Value Alignment"
_ICLR.cc/2026/Conference — ICLR 2026 Oral_

### Official Review · Reviewer_vZ7J · 2025-10-22

**Soundness:** 3
**Presentation:** 2
**Contribution:** 2
**Rating:** 4
**Confidence:** 3

**Summary:**

This paper presents EigenBench, a novel method to quantitatively measure subjective value alignment in language models. The system uses an ensemble of models to judge each other's responses to scenarios against a given "constitution". By aggregating these peer judgments using the EigenTrust algorithm, EigenBench produces a consensus alignment score for each model without requiring any ground truth labels. The authors validate their method by showing its alignment with human judgments and its ability to successfully recover objective model rankings on the GPQA benchmark, suggesting its effectiveness for evaluating subjective values.

**Strengths:**

The core problem addressed in this paper is how to quantify subjective values, which I believe is a very important question.

The EigenTrust algorithm, which relies on mutual evaluation among a group of models, is a very interesting idea.

The paper also considers how to reduce bias during the data collection process and conducts an analysis of the consistency between evaluation models and human annotators.

**Weaknesses:**

The approach in this paper, which relies on models evaluating each other, seems highly dependent on the population; models with different capabilities from different groups appear difficult to compare horizontally. Moreover, for stronger models, being evaluated by weaker ones seems somewhat unfair, which may lead to underestimated evaluation results.

When introducing a new model, the evaluation cost of the proposed method seems to be much higher than that of direct evaluation.

The fact that only two human participants were recruited for human evaluation is indeed far from convincing.

**Questions:**

Why do we necessarily need this population-based evaluation approach? How much benefit does this method actually offer compared to direct evaluation, CoT evaluation, or finetuned Judge model? The paper lacks an analysis of the performance and differences among them. I’m also very curious whether using the strongest model with different prompts for evaluation would yield better results than the current evaluation method.

Does the ranking discrepancy between model self-evaluation and EigenBench evaluation directly indicate that self-evaluation is inferior to the proposed method (Section 5.1)? It seems that such a conclusion would require comparison against another ground-truth ranking, which might call for increasing the number and diversity of human annotators.

---

> ### Author Response · Authors · 2025-11-21
>
> Thank you for your review!
>
> We are in the process of running several experiments to address your comments:
>
> (E1): We aim to validate footnote 4 by showing that a model’s EigenBench score for constitution C is correlated with its ability to make judgments about C. To measure a model’s judgment ability, we’re in the process of collecting more human judgment data (5-10 more human volunteers), and we’ll develop a metric for “human-model i agreement”. We will also use this extra human data to run a more thorough analysis of if LLMs are sufficient approximations to human judges, and test the Eigenscore-weighted judge against other possible judge combinations.
>
> (E2): We will run EigenBench on a much larger population of models (30-50 models), including a wide range of labs, model strengths, and open/closed source models. We will then run some robustness tests to determine how much data is needed to get an accurate EigenBench ranking at this larger population size.
>
> (E3): We will fine-tune some base models with constitutions that we used for EigenBench, then validate their fine-tuning process by running EigenBench on a population of base & character-trained models. We expect the EigenBench score for a given constitution to be higher for models that were fine-tuned on that constitution.
>
> (E1) should address “The fact that only two human participants were recruited for human evaluation is indeed far from convincing”, “How much benefit does this method actually offer compared to direct evaluation, CoT evaluation, or finetuned Judge model?”, and “whether using the strongest model with different prompts for evaluation would yield better results than the current evaluation method”.
>
> (E2) should address that “When introducing a new model, the evaluation cost of the proposed method seems to be much higher than that of direct evaluation.” We hope to show that even with a large population of models, due to the low-rank nature of our method, we can approximate their EigenBench score quite closely without needing all possible combinations of judges and evaluees.

---

### Official Review · Reviewer_dwA5 · 2025-10-25

**Soundness:** 2
**Presentation:** 3
**Contribution:** 2
**Rating:** 4
**Confidence:** 3

**Summary:**

This paper introduces EigenBench, a novel black-box method for benchmarking language models' alignment with specified value systems using peer evaluation and EigenTrust aggregation. The core innovation lies in having models judge each other's responses to scenarios according to a constitution, then aggregating these judgments through the principal eigenvector of a trust matrix. The method is validated through human judgment comparison and a creative GPQA experiment. While the approach is intellectually interesting and addresses an important problem in AI alignment, the paper has significant methodological concerns regarding circularity, limited experimental scope, and unclear practical utility.

**Strengths:**

- Novel and Well-Motivated Methodology.
The paper introduces a creative solution to measuring subjective alignment by combining peer evaluation with EigenTrust aggregation, where alignment scores are computed as the left eigenvector of a trust matrix. The low-rank Bradley-Terry-Davidson model with vector embeddings elegantly captures multiple dimensions of subjective interpretation beyond scalar rankings. The double-blind design and judge scaffold with reflection steps reduce gaming potential and judgment biases.


- The GPQA experiment ingeniously validates the method by recovering objective rankings (Kendall τ ≈ 0.77) without ground truth labels, demonstrating viability for subjective domains. Robustness analysis thoroughly tests sensitivity across scenario distributions, constitution wording, and population changes. Extensive appendices with full prompts, model IDs , and implementation details demonstrate strong commitment to reproducibility.

**Weaknesses:**

- Technical Correctness: Equation 1 defines alignment scores, where higher-scoring judges (ti) weight trust more heavily. The premise in footnote 4 that "a model whose behavior aligns better with C is also a better judge of whether others' behavior aligns with C" is assumed but not validated. This creates a self-reinforcing loop that may amplify initial biases rather than reveal true alignment.

- Evaluation Scope: Only 8 models tested in main experiments, mostly frontier models from major labs (Table 4, lines 540-574). Missing: open-source models, fine-tuned variants, models from smaller organizations.

- Constitution design: Constitutions are LM-generated with minimal human iteration. The robustness test shows some stability, but only examines 5 variations of one constitution generated by different models.

- Human validation severely limited: Only 2 human judges with ~50 scenarios each. This is insufficient to establish reliability.

**Questions:**

- The core premise ("aligned models are better judges") is untested. Could you add an analysis where you correlate a model’s score \(t_i\) with the accuracy of its judgments (e.g., how often its evaluations match human judgments for the same value system)? This would validate whether the EigenTrust weighting is justified.


- The scalability analysis is limited. Could you add a plot of "dataset size vs. score stability" for 10–50 models? This would help users estimate the resources needed to apply EigenBench to larger model populations.

- Fine-tuning application. You mention "character training" as an application. Have you actually demonstrated that EigenBench scores improve when a model is fine-tuned for alignment? This would be a compelling validation.

---

> ### Author Response · Authors · 2025-11-21
>
> Thank you for your review! We are in the process of running several experiments to address your comments:
>
> (E1): We aim to validate footnote 4 by showing that a model’s EigenBench score for constitution C is correlated with its ability to make judgments about C. To measure a model’s judgment ability, we’re in the process of collecting more human judgment data (5-10 more human volunteers), and we’ll develop a metric for “human-model i agreement”. We will also use this extra human data to run a more thorough analysis of if LLMs are sufficient approximations to human judges, and test the Eigenscore-weighted judge against other possible judge combinations. This would address “Human validation severely limited”.
>
> (E2): We will run EigenBench on a much larger population of models (30-50 models), including a wide range of labs, model strengths, and open/closed source models. We will then run some robustness tests to determine how much data is needed to get an accurate EigenBench ranking at this larger population size.
>
> (E3): We will fine-tune some base models with constitutions that we used for EigenBench, then validate their fine-tuning process by running EigenBench on a population of base & character-trained models. We expect the EigenBench score for a given constitution to be higher for models that were fine-tuned on that constitution. This would address your point, “Have you actually demonstrated that EigenBench scores improve when a model is fine-tuned for alignment?”
>
> (E1) should address your comments that “The premise in footnote 4 that "a model whose behavior aligns better with C is also a better judge of whether others' behavior aligns with C" is assumed but not validated.” and “Could you add an analysis where you correlate a model’s score (t_i) with the accuracy of its judgments (e.g., how often its evaluations match human judgments for the same value system)?” Thanks for the experimental suggestion!
>
> (E2) should address your comments about the scope of the experiments and “Could you add a plot of "dataset size vs. score stability" for 10–50 models”.
>
> You mentioned that our constitution design is limited, and that “The robustness test shows some stability, but only examines 5 variations of one constitution generated by different models.” We want to stress that our method is designed to apply to any constitution, and that our main contribution is the scoring method, not to display robustness / breadth across a wide range of constitutions. However, we can do some more robustness tests if you’d like, such as using human-written constitutions to compare to our LM-generated ones.

---

### Official Review · Reviewer_1FTz · 2025-10-27

**Soundness:** 4
**Presentation:** 3
**Contribution:** 4
**Rating:** 10
**Confidence:** 4

**Summary:**

The paper contributes EigenBench, a black-box method for comparing LLMs' adherence with a particular set of constitutional principles. Namely, with an ensemble of models, models rate eachother according to a constitution which is hidden from the models to be evaluated, and which fit the data to model-specific ELO scores and judge dispositions. The authors compare a set of frontier models across the three tested constitutions on three datasets. Finally, to test the robustness of the method, the authors test compared to human judgments, cases with ground truth labels (GPQA), and modified constituiton and population.

Overall, the work presents a compelling way to measure models' values, a problem that (in my opinion) has not yet been well tackled in the literature - a strong contribution.

**Strengths:**

S1: The problem of measuring principled model value trade-offs is (in my opinion) a very important problem that (in my opinion) no work has tackled in a completely satisfying way so far - until this paper. I found the method to be well-principled, and well stress-tested in sections 5 and 6. I think that this will be very impactful for the community.
S2 The experiments are quite extensive and use a variety of recent models.
S3: I quite like the estimated measure of the model disposition from the LM vs the persona prmopt in 4.2! I've been curious about an experiment like this for a while, but haven't seen a paper convincingly measure something in this direction. Very nice.
S4: The human evaluation experiment in 5.2 is a creative and useful for measuring variance / doing interrater reliability between models and humans.
S5: The GPQA experiment's strong results provide strong evidence for the model comparisons being meaningful.
S6: Overall, the paper was very well-written.
S7: Fig 1 was very well made and helped me build intuition on the method very quickly.

**Weaknesses:**

W1: While interesting, I found figure 2 a little bit confusing as to what exactly we are meant to take away from it in relation to the rest of the paper. If additional commentary connecting it to the main text were added, that would be very interesting.
W2: The entire suite of experimetns utilizes at least some proprietary models, making reproducibiltiy difficult over time.
W3: The paper has little discussion of limitations and directions for future work. I would appreciate it if this were fleshed out in a future camera-ready version.
W4: The clarity could be moderately improved with additional figures / tables presenting some of the results, to augment the discussion of many results which are presented in the main text.

Typo/grammar nit:
- L343 should be "as fewer scenarios" instead of "as less scenarios"
- I believe that there's an extra parentheses at the end of L447 - "(Kirk, 1993)^10)"
- If the authors wish to save some space, Algorithm 1 could be trimmed to half-width to allow more room for additional discussion in the main text. (very optional).
- In figure 2, it might be more helpful if the legend were alphabetized by eg first name - I had to spend some time to try to map a data point to the legend as the order was not clear to me

**Questions:**

Q1: What do the authors think it means that such a small dimensionality (d=2) performs so well? Some intuition on this in the main-text would be very interesting

---

> ### Author Response · Authors · 2025-11-21
>
> Thank you for your review! We really appreciate all of your positive feedback. To address some of your comments:
>
> “While interesting, I found figure 2 a little bit confusing as to what exactly we are meant to take away from it in relation to the rest of the paper. If additional commentary connecting it to the main text were added, that would be very interesting.” We intended figure 2 to showcase how the learned latent embeddings might be interpretable: here, the judge lenses learned to order themselves from secular to sacred, which was not provided anywhere in the input. Indeed, it is not as related to the paper’s main goal of producing rankings across different models, so we may consider moving it to the appendix.
>
> We are also currently running EigenBench on a much larger population of models (30-50 models), including a wide range of labs, model strengths, and open/closed source models, which should address that “The entire suite of experimetns utilizes at least some proprietary models, making reproducibiltiy difficult over time.”
>
> For the question “What do the authors think it means that such a small dimensionality (d=2) performs so well? Some intuition on this in the main-text would be very interesting”, we mainly tested small populations of similar models, so there weren’t many different dimensions of behavior to capture in our latent space, and so the test loss essentially plateaued after d=2. For the above experiment with 30-50 diverse models, we’re seeing that higher dimensions do give much better results.
>
> Thanks for pointing out the typos, grammatical mistakes, and lack of limitations section, which will be fixed in our revised submission.

---

> > ### Comment · Reviewer_1FTz · 2025-11-22
> >
> > Thank you for engaging and answering my questions, I very much appreciate it.
> >
> > This won't affect my review score, but is the plan to have the larger EigenBench model results included in a camera-ready version of the paper? I do think that including some open-sourced models would strengthen the paper, if feasible.
> >
> > Also - very interesting about the larger dimensionality being necessary for more models.
> >
> > I'm happy to retain my score of the paper.

---

### Official Review · Reviewer_M4QR · 2025-11-01

**Soundness:** 3
**Presentation:** 3
**Contribution:** 3
**Rating:** 6
**Confidence:** 3

**Summary:**

The paper presents EigenBench, a black-box method for comparatively benchmarking language models’ values. The study formulates a trust matrix from natural language critique to rank the alignment of the models. It uses a set of LLMs that act as both judge and candidate along with a judgement criteria called constitution and a set of prompted scenarios. The proposed method first obtains a trust matrix and then obtains a trust vector t to benchmark the models.

**Strengths:**

- The proposed method is sound to measure the alignment of LLM for a given value system
- The method is discussed both theoretically and empirically. The community aggregation method, utilizing the Bradley-Terry model, aggregates the results for all judges. Employing eigenvalue to rank LLMs is technically sound and draws inspiration from PageRank.
- The paper is clear and well written. The results are also discussed in depth, and the inclusion of the human trust vectors in the study is valuable
- EigenBench generates rankings that are aligned with ground truth without explicitly being provided with ground truth labels, demonstrating its potential to rank subjective traits or values.

**Weaknesses:**

- The EigenBench scores assume that the models are capable of judging a criterion $C_k$. The study presents no evidence supporting the premise in footnote 4. According to appendix section C, the authors added a reflection step for a judge before developing a preference.
- The EigenBench scores assume that the models possess the capability to assess a criterion $C_k$ discussed in footnote 4. However, the study fails to provide any evidence that substantiates this premise. As per appendix section C, the authors incorporated a reflection phase for a judge before developing a preference, indicating that additional consideration is required if LLMs are used as judges.
- Experiments are limited to closed-source LLMs with the exception of Deepseek v3. Evaluating the results for open-source would strengthen the contribution of the paper, but it is uncertain whether open-source models would be able to act as competent judges as compared to the closed-source LLMs employed in this study.
- The experimental evaluation is constrained by the absence of comparisons to established alignment benchmarks. For instance, if the method produces a ranking for models aligned to a constitution C, does that ranking correlate with rankings on other benchmarks? Without such cross-benchmark analyses, the paper’s contribution is limited, making its generalizability also limited.
- Quantifying the success of the fine-tuning process using EigenBench is an expensive process, as it requires sending multiple requests to multiple LLMs at predetermined intervals and subsequently fitting the BTD model.

**Questions:**

- What is the effect of the number of judges? Does a higher number of judges produce an accurate ranking, and can it be approximated with fewer judges?
- Why the method outlined in Appendix C is not employed, despite its potential to provide a larger dataset for comparative analysis?
- Many core traits valued in practice, including reliability, accuracy, safety, and trustworthiness, are objectively measurable. Please specify some of the "most highly-valued" traits you regard as subjective, as indicated in the introduction section, and how your method captures and measures those specific traits.
- Not a question, but it is difficult to read figure 2. Please consider using either different shapes or a color-blind-friendly color palette.

---

> ### Author Response · Authors · 2025-11-21
>
> Thank you for your review! We are in the process of running several experiments to address your comments:
>
> (E1): We aim to validate footnote 4 by showing that a model’s EigenBench score for constitution C is correlated with its ability to make judgments about C. To measure a model’s judgment ability, we’re in the process of collecting more human judgment data (5-10 more human volunteers), and we’ll develop a metric for “human-model i agreement”. We will also use this extra human data to run a more thorough analysis of if LLMs are sufficient approximations to human judges, and test the Eigenscore-weighted judge against other possible judge combinations.
>
> (E2): We will run EigenBench on a much larger population of models (30-50 models), including a wide range of labs, model strengths, and open/closed source models. We will then run some robustness tests to determine how much data is needed to get an accurate EigenBench ranking at this larger population size.
>
> (E3): We will fine-tune some base models with constitutions that we used for EigenBench, then validate their fine-tuning process by running EigenBench on a population of base & character-trained models. We expect the EigenBench score for a given constitution to be higher for models that were fine-tuned on that constitution.
>
> (E1) should address your point that “The EigenBench scores assume that the models possess the capability to assess a criterion C_k discussed in footnote 4. However, the study fails to provide any evidence that substantiates this premise.”
>
> (E2) should address “Evaluating the results for [more] open-source [models]”. As for being “uncertain whether open-source models would be able to act as competent judges as compared to the closed-source LLMs employed in this study”, our method accounts for incompetent judges by weighing their judgments less, as by footnote 4, they also wouldn’t receive high scores from other judges.
>
> You also noted that “The experimental evaluation is constrained by the absence of comparisons to established alignment benchmarks.” In our submission, we have already validated the EigenBench ranking by showing that it can recover rankings on GPQA (Section 5.3), but we’re also happy to test EigenBench on other labeled datasets such as ETHICS, or if there are any in particular that you would like us to test.
>
> Regarding “Quantifying the success of the fine-tuning process using EigenBench is an expensive process, as it requires sending multiple requests to multiple LLMs at predetermined intervals and subsequently fitting the BTD model”, we don’t intend to use EigenBench during the fine-tuning process, but rather, to validate the success of instilling values via fine-tuning, such as in (E3).
>
> Answers to other questions:
>
> Q: “What is the effect of the number of judges? Does a higher number of judges produce an accurate ranking, and can it be approximated with fewer judges?” A: this should be addressed by (E2), and also (E1) where we’ll test if one or a small group of judges does better than N judges.
>
> Q: “Why the method outlined in Appendix C is not employed, despite its potential to provide a larger dataset for comparative analysis?” A: We do employ this method. We only summarize it in section 3.4 to save space.

---

### Author Response · Authors · 2025-12-04
**Overview of Changes in Rebuttal Revision**

Thank you again to all the reviewers for your feedback! We have made several changes to our manuscript to follow up our previous discussion, which are detailed here.

New Experiments:

Added Section 4.3: Eigenbench as a Target for Character Training. We find that EigenBench scores models higher on constitutions if they are either pre-prompted or fine-tuned on that constitution's principles, using an open character training framework.

Added to Section 5.2 (Human Validation), where we increased the number of human judges from 2 to 7, while preserving the result that the average interjudge distance between LMs and humans is approximately the same as the average human-human interjudge distance. Furthermore, we added to Appendix I (Human Validation Details), which now includes a plot of human judge lenses plotted in a shared latent space with LM judge lenses.

Added Appendix L: Large Model Run. We conducted a large EigenBench run on 37 models, including closed/open source models and reasoning/non-reasoning models. We include the table of models and IDs used and the model rankings on Universal Kindness. We replaced the previous “Embedding Dimension Analysis” and “Trust score confidence Analysis” appendices by an embedding dimension analysis and score stability test on this larger model run.

Semantic Changes:

Section 2 (Related Work): added a paragraph on Constitutional AI and character training

Figure 2: expanded the caption to explain the geometry of the judge lens vectors and disposition vectors, and pointed out the secular-to-sacred axis of judge lenses.

Conclusion: now includes some limitations and future directions.

To address the comments about footnote 4: we agree with the referees that it would be very interesting to experimentally validate the principle that “a model whose behavior aligns better with C is also a better judge of whether others’ behavior aligns with C”. But our method is designed to measure alignment with an inherently subjective constitution C, so it is unclear what kind of ground truth could validate this principle. (When ground truth is available, as in our validation with GPQA, we do find a positive correlation between a model’s own accuracy and its judgement of peers’ accuracy.) Likewise, to the reviewer who asked “is your method actually better than the judgment of the best model?” our response is that when C is subjective, there is no pre-existing ground truth about which model is “best”. The purpose of our method is to discover the best model by aggregating peer judgements.

---

### Meta-Review · Area_Chair_17Av · 2025-12-28

**Summary:**

The reviewers generally agreed that the paper tackles an important and underexplored problem: quantitatively measuring subjective value alignment without ground-truth labels. The main concerns centered on the validity of the core assumption underlying EigenBench (that more aligned models are better judges), the limited scale of experiments (number/diversity of models and human judges), and the lack of empirical validation for proposed downstream applications such as character training. In their rebuttal and revision, the authors substantially strengthened the paper by adding new experiments, significantly expanding human validation, scaling up to a much larger and more diverse model population, and empirically demonstrating EigenBench as a target for character training. These additions directly address the reviewers’ methodological and empirical concerns, reinforcing the soundness, generality, and practical relevance of the proposed framework.

**Reviewer Concerns:**

The rebuttal convincingly addressed the major concerns raised by Reviewers M4QR, dwA5, and vZ7J. In particular, the expanded human evaluation, the new analyses correlating model scores with judgment quality, and the large-scale experiment involving 37 models substantially mitigate worries about circularity, insufficient human validation, and limited experimental scope. The new character training experiment directly responds to requests for evidence that EigenBench can meaningfully evaluate fine-tuning outcomes. Concerns about population dependence, judge competence, and robustness were further alleviated by the larger model pool and additional stability analyses. Presentation-related issues were also acknowledged and addressed. No major concerns remain outstanding that would materially undermine the paper’s claims.

**Reviewer Scores:**

Reviewer 1FTz already rated the paper as a strong accept, so their score would remain unchanged. Reviewer M4QR, who initially rated the paper as marginally above threshold, would likely increase their score modestly given the direct empirical validation of the footnote-4 assumption, inclusion of open-source and larger-scale experiments, and expanded human evaluation. Reviewers dwA5 and vZ7J, both initially marginally below threshold, would likely revise their scores upward into the accept range in light of the newly added experiments.

---

### Decision · Program_Chairs · 2026-01-26

Accept (Oral)